# HIV testing and linkage to care—A case of a mobile diagnostic and counseling service in Mbeya, Tanzania; A quantitative study

Amani Kway[1]*, Issa Sabi[1], Willyhelmina Olomi[1‡], Ruby Doryn Mcharo[1], Erica Sanga[2], Wiston William[1‡], Ombeni Chimbe[1], Nyanda Elias Ntinginya[1], Lucas Maganga[1]

1 National Institute for Medical Research - Mbeya Medical Research Centre, Mbeya, Tanzania, 2 National Institute for Medical Research - Mwanza Centre, Mwanza, Tanzania

☯ These authors contributed equally to this work.
‡ WO and WW also contributed equally to this work.
* akway@nimr-mmrc.org

## Abstract

HIV-care programmes are faced with significant challenges in getting newly diagnosed People Living with Human Immunodeficiency Virus (PLHIV) linked to care despite massive investment in HIV prevention, treatment and care. This study assessed the performance of mobile HIV Testing and Counseling service (mHTC) in provision of HIV-testing and linkage to care of newly diagnosed PLHIV from Key and Vulnerable Populations (KVPs). A retrospective review of the records of 25,248 clients was extracted from the mHTC database from October-2016 to September-2018. Of 25,248 clients, 51.71% were in 25–45 years age group, 55.4% were males, 60.5% were married and 62.1% had primary level of education. The median age of clients was 31 (IQR: 23–42) years. Out of the clients tested, 800 (3.17%) were diagnosed HIV-positive. Positivity was high among females 450 (4%), age group 25–45 years 538 (4.12%), divorced 202 (7.41%) and clients with primary level of education 504 (3.21%). An association between HIV status and sex, age group, relationship status and level of education was observed (P<0001). Out of the 800 HIV-positive clients, 418 (52.30%) were successfully linked to care. Among the positive clients, 5/6 (83.33%) children below 15 years old, 238/450 (52.89%) females and 39/64 (60.94%) widows were successfully linked to care. In the multivariable log binomial regression model age of the clients was associated with successful linkage to care. The mHTC was able to reach KVP clients; overall linkage for both sexes was 52.30% below the recommended UNAIDS 90-90-90 target. Raising the need to address the challenges associated with linkage and specific care for KVPs as a subset of the general population. The mHTC has shown that it is feasible to improve the reach of KVP clients; however, further research is required to examine the quality of this service at the community level.

**Data Availability Statement:** The datasets used and/or analyzed during the current study are uploaded as supplementary information.

**Funding:** AK was funded by the NIMR-MMRC through the institution seed grant project for young scientists. The views expressed in this paper are those of the authors and do not necessarily represent the views of the funding agency. The funders had no role in study design, data collection and analysis, decision to publish, or preparation of the manuscript.

**Competing interests:** The authors have declared that no competing interests exist.

## Introduction

HIV-care programmes are faced with significant challenges in getting newly diagnosed People Living with Human Immunodeficiency Virus (PLHIV) linked to care and retaining those already on Highly Active Anti-Retroviral Therapy (HAART). Data indicates that, only 53% of PLHIV in 2015 were on HAART in Sub-Saharan Africa (SSA) [1]. When compared to other areas, SSA has low rates of linkage to care after HIV-testing, with about two-thirds of patients Lost To Follow Up (LTFU) before initiation of HAART [2, 3]. Innovative strategies to facilitate linkage in to HIV-care in SSA are thus needed, especially as the number of people in need of HAART grows while countries continue to roll out the World Health Organization (WHO) "test and treat" recommendations throughout the region. Frequently cited barriers to linkage include; psychosocial like stigma and fear of disclosure; economic such as inability to afford transport costs, distance to health facilities and health system factors [4]. Attrition between testing and HAART initiation occurs because of the clients tested positive, a few are not successfully linked to care and therefore contribute to the spread of new HIV infections.

In the efforts to curb the HIV epidemic in the Southern Highlands regions of Tanzania, the National Institute for Medical Research-Mbeya Medical Research Centre (NIMR-MMRC) mobile laboratory has been implementing a "mobile HIV Testing and Counseling service (mHTC)" with funds from PEPFAR since 2009. Visiting between 8–12 Key and Vulnerable Populations (KVPs) sites every 3 months in Mbeya. mHTC was coupled with health education and screening for Sexually Transmitted Infection (STIs), Tuberculosis as well as cervical cancer.

Despite massive investments in HIV prevention, treatment and care, HIV/AIDS remains a substantial cause of morbidity and mortality in this region. Optimal design of effective approaches of linkage to HIV-care is hampered by limited understanding of factors beyond individual behavioral determinants (e.g. knowledge, self-efficacy, income) that mediate engagement in care [5].

In Tanzania 52.2% of PLHIV 15 to 64 years, were aware of their HIV status and 90.9% of those who knew their status were on HAART [6]; more innovative and client friendly modalities for HIV-testing and early initiation of therapy are needed in order to meet the United Nations Programme on HIV/AIDS (UNAIDS) 90-90-90 targets. Key populations are special groups characterized by specific higher-risk behaviors, posing them at increased risk of HIV regardless of the local context. Comprising of men who have sex with men, people who inject drugs, people in prisons and other closed settings, sex workers and transgender people [7]. Vulnerable populations are groups of people who are more prone to HIV infection in certain contexts, such as adolescents girls, mobile workers etc [7]. Among the important factors driving the HIV epidemic in Mbeya are high population mobility associated with cross-border socio-economic activities, KVPs such as Female Sex Workers (FSWs) and their clients, miners, long distance truck drivers, poverty and inadequate health facilities that can be accessed by KVPs. The KVPs are a special group contributing to the burden of HIV in the region. The observed National AIDS Control Program (NACP) program performance data as per THIS 2016–2017 is likely to be different for KVPs. This study assessed the performance of mHTC services in provision of HIV-testing and linkage to care of newly diagnosed PLHIV from KVPs.

## Methods

A retrospective review of the routine collected mHTC data was conducted between October-2016 to September-2018.

This study was conducted in rural areas of Mbeya region in the Southwestern highlands of Tanzania. The HIV prevalence for the whole region is 9.3% ranking third as having the highest

HIV prevalence in Tanzania [6]. The Region has 2,707,410 Population as per national census of 2012 [8]. The research work was a KVP focused study although General Populations (GPs) in need of the services were not denied for the reasons of the existing interactions between them. The targeted population was physically reached by mHTC and linkage to care of each newly diagnosed HIV positive client was done through physical escort to CTC of their choice. The approach of mHTC services started with provision of HIV-testing and ended with linkage to care of newly diagnosed PLHIV from KVPs.

## Data collection

Records of 25248 individuals from KVPs who received HIV-test through mHTC between October-2016 to September-2018 were extracted from the database. KVPs who had taken HIV-test within one month were regarded as being aware of their status and did not receive the test. mHTC has been implementing mobile HIV testing covering between 8–12 KVP sites every 3 months in Mbeya region. Ethical clearance was sought from Mbeya Medical Research and Ethics review Committee (MMREC). In a letter with reference number: SZEC-2439/R.A/V.1/31.

Permission to access the mHTC database was requested verbally from the administration of NIMR-MMRC and it was granted. However, the dataset did not include individual identifiers hence anonymity and confidentiality were observed.

## Variables

The independent variables of interest in the analysis included age, sex, relationship status and level of education. HIV sero-status and linkage status were used as outcome variables. Linkage was defined as the confirmation of HIV-positive diagnosis and first HIV-specific clinic visit; as per WHO [9]. Employing the "test and treat strategy", newly diagnosed PLHIV were Physically linked to Care and Treatment Center (CTC) of their choice by the mHTC team on the same day.

## Data analysis

Data were entered into Microsoft Access database and then imported for analysis into Stata software version 15. Descriptive statistics were generated. Continuous variables were summarized using means and standard deviation as well as medians and interquartile ranges. Dichotomous variables were summarized using frequencies and percentages. Chi-square tests were used to evaluate bivariate associations between baseline variables and HIV-positivity. Log-binomial regression model was used to identify demographic factors associated with linkage to CTCs.

## Results

Between October-2016 to September-2018, the mHTC attended 25,248 clients from the KVP communities in the region of Mbeya. The majority of the clients were in the age group 25–45 (51.71%). The median age of the visited clients was 31 (IQR: 23–42) years. Most of the clients were males 55.4% and married 60.5%. Two-thirds of the clients attended had primary level of education (Table 1).

Of the 25,248 clients who tested for HIV, 800 (3.17%) were diagnosed HIV-positive. Positivity was high for female clients 450 (4%), older age group 25–45 years 538 (4.12%), divorced 202 (7.41%), and with primary level of education 504 (3.21%) compared to their counterparts. We observed a strong association between HIV status and sex, age group, relationship status and level of education (all p<0.0001, chi-square test) (Table 2).

**Table 1. Social-demographic characteristics of Key and Vulnerable Populations.**

| Social-demographic factors | | |
|---|---|---|
| | **Frequency** | **Percentage (%)** |
| **Age Median(IQR)** | 31(23–42) | |
| **Age groups** | | |
| <15 | 474 | 1.88 |
| 15–24 | 7,034 | 27.86 |
| 25–45 | 13,056 | 51.71 |
| >45 | 4,684 | 18.55 |
| **Sex** | | |
| Male | 13,997 | 55.44 |
| Female | 11,251 | 44.56 |
| **Relationship status** | | |
| Single | 5,704 | 22.59 |
| Married | 15,273 | 60.49 |
| Divorced | 2,727 | 10.80 |
| Widowed | 1,077 | 4.27 |
| Cohabiting | 58 | 0.23 |
| Non applicable * | 409 | 1.62 |
| **Level of education** | | |
| No formal education | 4,110 | 16.28 |
| Primary education | 15,687 | 62.13 |
| Secondary education | 2,787 | 11.04 |
| College | 489 | 1.94 |
| Missing information | 2,175 | 8.61 |
| **Year** | | |
| 2016 | 2175 | 8.60 |
| 2017 | 15002 | 59.40 |
| 2018 | 8071 | 32.00 |
| **Total** | **25,248** | **100.00** |

*Non applicable represents clients <15 years who had not yet started relationships

## Linkage to care

Out of 800 clients diagnosed HIV-positive, only 418 (52.25%) were successfully linked to care. Descriptive analyses revealed that of the diagnosed HIV-positive clients linked to-care 5/6 (83.33%) were children below 15 years, 238/450 (52.89%) were females, 39/64 (60.94%) were widows. Results from multivariable log binomial regression indicated that age of the clients was associated with successful linkage to care (Table 3).

## Discussion

In the period under investigation, the mHTC demonstrated an HIV-positivity of 3.17% in this rural setting. Our findings are closely comparable with the national data from the NACP published in 2017, indicating 4.3% prevalence for rural and 7.2% for urban areas [10]. However, it is likely that there is significant variation in the regional data for the GP and KVPs in the rural areas of Tanzania. Among the important factors driving the HIV epidemic in Mbeya are high population mobility associated with cross-border socio-economic activities, KVPs such as FSWs, miners, long distance truck drivers, poverty and inadequate health

**Table 2. HIV sero-status among tested individuals.**

| Social-demographic factors | HIV sero-status of tested individuals n = 25,248 | | Total n | p values |
|---|---|---|---|---|
| | n (%) | n (%) | n | |
| | HIV+ = 800 (3.17) | HIV- = 24,448 (96.83) | | |
| **Age groups** | | | | |
| <15 | 6 (1.27) | 468 (98.73) | 474 | P<0.0001 |
| 15–24 | 171 (2.43) | 6,863 (97.57) | 7,034 | |
| 25–45 | 538 (4.12) | 12,518 (95.88) | 13,056 | |
| >45 | 85 (1.81) | 4,599 (98.19) | 4,684 | |
| **Sex** | | | | |
| Male | 350 (2.50) | 13,647 (97.50) | 13,997 | P<0.0001 |
| Female | 450 (4.00) | 10,801 (96.00) | 11,251 | |
| **Relationship status** | | | | |
| Single | 151 (2.65) | 5,553 (97.35) | 5,704 | P<0.0001 |
| Married | 378 (2.47) | 14,895 (97.53) | 15,273 | |
| Divorced | 202 (7.41) | 2,525 (92.59) | 2,727 | |
| Widowed | 64 (5.94) | 1,013 (94.06) | 1,077 | |
| Cohabiting | 1 (1.72) | 57 (98.28) | 58 | |
| Non applicable* | 4 (0.98) | 405 (99.02) | 409 | |
| **Level of education** | | | | |
| No formal education | 128 (3.11) | 3,982 (96.89) | 4,110 | P<0.0001 |
| Primary education | 504 (3.21) | 15,183 (96.79) | 15,687 | |
| Secondary education | 61 (2.19) | 2,726 (97.81) | 2,787 | |
| College | 1 (0.20) | 488 (99.80) | 489 | |
| Missing information | 106 (4.87) | 2,069 (95.13) | 2,175 | |
| **Year** | | | | |
| 2016 | 105 (4.80) | 2070 (95.20) | 2175 | P<0.0001 |
| 2017 | 441 (2.90) | 14561 (97.10) | 15002 | |
| 2018 | 253 (3.10) | 7818 (96.90) | 8071 | |

*Non applicable represents clients <15 years who had not yet started relationships

facilities that can be accessed by KVPs. The KVPs are a special group contributing to the burden of HIV in the region. The observed NACP program performance data as per THIS 2016–2017 is likely to be different for KVPs. Our findings showed that HIV prevalence among KVPs in rural areas is lower as compared to the national prevalence in the same period, the observed difference could be explained by the fact that; although mHTC was targeting KVP clients but also GP who needed this service were not excluded upon request and therefore diluting the data on HIV prevalence; The limited data available shows close similarity with findings from another study in Mbeya, that indicated lower HIV infections in rural areas as compared to urban [11]. Another study among randomly selected household members in rural areas in Kilimanjaro indicated that HIV prevalence in the targeted communities ranged from 1.4–4.5% [12].

About half (52.30%) of the diagnosed PLHIV were successfully linked to care; this is much lower than the 90% recommended by UNAIDS. The lower linkage data could be possibly attributed by the underlying stigma and discrimination towards disclosure of HIV status that exist in our communities. The fact that mHTC was largely targeting clients from the KVP communities who are known to be highly mobile, linkage modalities and HIV care

**Table 3. Factors associated with linkage to care among clients diagnosed HIV-positive.**

| Social-demographic factors | Diagnosed HIV-positive clients n = 800 | | Total (N) | | |
|---|---|---|---|---|---|
| | Clients successfully linked to care = 418 (52.30%) | Clients unsuccessfully linked to care = 382 (47.70%) | | | |
| | n (%) | n (%) | | CPR (95% CI) | APR (95% CI) |
| **Age groups** | | | | | |
| 15–24 | 77 (45.03) | 94 (54.97) | 171 | Ref[1] | Ref |
| <15 | 5 (83.33) | 1 (16.67) | 6 | 1.85 (1.25–2.75) | 1.63(1.06–2.49)*[2] |
| 25–45 | 287 (53.35) | 251 (46.65) | 538 | 1.18 (0.98–1.42) | 1.24 (1.03–1.49)* |
| >45 | 49 (57.65) | 36 (42.35) | 85 | 1.28 (1.01–1.64) | 1.33 (1.03–1.70)* |
| **Sex** | | | | | |
| Male | 180 (51.43) | 170 (48.57) | 350 | Ref | Ref |
| Female | 238 (52.89) | 212 (47.11) | 450 | 1.03 (0.90–1.18) | 1.04 (0.91–1.19) |
| **Relationship status** | | | | | |
| Single | 77 (50.99) | 74 (49.01) | 151 | Ref | |
| Married | 189 (50.00) | 189 (50.00) | 378 | 0.98 (0.81–1.81) | |
| Divorced | 109 (53.96) | 93 (46.04) | 202 | 1.06 (0.86–1.29) | |
| Widowed | 39 (60.94) | 25 (39.06) | 64 | 1.19 (0.93–1.54) | |
| **Level of education** | | | | | |
| No formal education | 68 (53.13) | 60 (46.88) | 128 | Ref | |
| Primary education | 252 (50.00) | 252 (50.00) | 504 | 0.94 (0.78–1.13) | |
| Secondary education | 32 (52.46) | 29 (47.54) | 61 | 0.98 (0.74–1.32) | |
| **Year** | | | | | |
| 2016 | 64 (61.00) | 41 (39.10) | 105 | Ref | Ref |
| 2017 | 199 (45.10) | 242 (54.90) | 441 | 0.74 (0.61–0.89) | 0.73 (0.61–0.88)* |
| 2018 | 154 (60.90) | 99 (39.10) | 253 | 0.99 (0.83–1.20) | 0.98 (0.82–1.18) |

[1] Ref-Reference group,

[2]* $p < 0.05$ * CPR- Crude Prevalence Ratio, APR- Adjusted Prevalence Ratio, Multivariable model adjust for Age group, Level of education, Sex and Relationship status

services that are applicable to the GP may not be feasible for this category of population as they tend to not very well interact within the routine CTC settings. Poor linkage is one of the predominant challenges against HIV prevention efforts. Literature indicates that, Clients are registering for CTC services far away from their residential places because they don't want to be identified by their fellows as HIV-positive [12]. Furthermore, the risks for poor linkage such as; mobility, stigma and discrimination, religious and superstition beliefs, hopelessness as well as poverty are also associated with defaulting from care raising the number of LTFU [12, 13].

mHTC had mostly reached clients in the age group of 25–45 years (51.71%) followed by 15–24 years (27.86%); the age group 25–45 years had higher HIV-prevalence (4.12%)

followed by 15–24 years (2.43%). Although THIS 2016–2017 indicated that percentage of HIV-positivity was highest among the age group of 15–24 years [6], our findings may be explained by the fact that majority of our clients were in the age group of 25–45 years a common phenomenon for KVPs in rural areas. Successful linkage to care was a challenge among clients in the age group of 15–24 and 25–45 years (45.03% and 57.65%) versus the age group >45 years (57.65%), a similar finding to other literatures in Tanzania [6, 12]. This data calls for more efforts to be directed towards not only the age groups of 15–24 and 25–45 years but also the age group >45 years as they also contribute largely to the HIV prevalence and new infections in the rural settings.

The mHTC has demonstrated high coverage for men (55.4%), through following them directly to their workplaces, recreational facilities and moonlight services, complementing the current national campaign of 'Mwanaume Jitambue' which aims at reaching more men with HIV Testing Services (HTS). In Tanzania like most African countries, the indirect opportunity costs associated with temporary suspension of economic activities and production has shown to discourage the uptake of HTS [14–16]. The deployment of such services like mHTC may possibly contribute to improvement of male coverage. Females were observed to have more HIV infections (4%) compared to males (2.5%), this data is consistent with THIS 2016–2017 [6]. The observed data may be explained by the fact that most of the females reached by mHTC were FSWs compared to males who mostly were either miners, fishermen or long-distance truck drivers. Despite mHTC having good coverage for males than females on HIV-testing; males had slightly lower percentage of successful linkage (51.43%) compared to females (52.89%) in line with multiple literatures [6, 12, 14–16].

Moreover, significant variation in percentage of HIV-positivity was observed with relationship status, whereby divorced clients had the highest (7.41%), followed by widows (5.94%), this is possibly explained by the fact that all of them were women with no men to support them, so they tend to engage themselves in high-risk activities making them more vulnerable to HIV. Successful linkage varied with relationship status among the clients with widows having more linkage (60.94%) followed by divorced (53.96%), while married were the lowest (50.00%). This finding could possibly be explained by the fact that widows unlike the married women, do not need to request permission for linkage from their partners. In Tanzania like many other African countries; men appreciate decision making autonomy on HTS [15], meanwhile women lack control of taking HIV-test and they have to discuss and obtain permission from their sexual partners [15, 17]. Furthermore requesting for consent raised an impression of infidelity [17] worse enough those diagnosed to be HIV-positive risked being stigmatized for contracting the infection [15]. The mHTC can be used as a point of entry for HTS of women through starting with their male sexual partners whom they have a good coverage; as basically decision on seeking health care lies with men [15], moreover support from spouses enhance uptake of HTS in females [18]. Moreover, overcoming the fear of; social isolation [14, 16, 19–21], losing intimate partners [16, 20–22] and marital instability comprising of physical abuse, abandonment/divorce [23, 24] will have an added advantage of increasing the number of clients up take of HIV-test.

mHTC was effective in reaching clients with low literacy level; 62.13% had primary and 16.28% no any level of formal education. Individuals with primary level of education had the highest percentage of HIV-positivity (3.21%) followed by those with no formal education (3.11%); in line with findings of THIS 2016–2017 [6]. It was also observed that their linkage to care is pretty challenging compared to those with higher literacy levels from secondary school and above who tend to comply easily with linkage, this observation was similar to the findings from the THIS 2016–2017 [6].

## Limitations

This study used retrospective data collected by the mHTC, some information such as factors affecting linkage were not captured i.e., stigma, discrimination, poverty, distance to the health facilities, while several other information like level of education had some missing data. The missing data is likely to have contributed to the findings observed in this study. Due to the nature of the mHTC, some diagnosed PLHIV who were linked successfully after the mHTC have left a specific catchment area, their data could not be captured in this analysis. This study did not capture data on STIs, Tuberculosis and cervical cancer.

## Conclusion

The mHTC was able to reach KVP clients with limited access to health care. Overall linkage for both sexes was 52.30% below the recommended UNAIDS target. This raises the need to address the challenges associated with linkage and specific care for KVPs as a subset of the GP. This study has shown that it is feasible to improve the reach of KVP clients with mHTC when coupled with other services; however, further research is required to examine the quality of this service at the community level.

## Supporting information

**S1 Data. HIV testing and linkage to care data.**
(XLSX)

## Acknowledgments

The authors acknowledge the regional health authorities of Mbeya for their tireless efforts to combat HIV infection and support PLHIV in their region.

## Author Contributions

**Conceptualization:** Amani Kway, Issa Sabi, Willyhelmina Olomi, Ruby Doryn Mcharo, Erica Sanga, Wiston William, Ombeni Chimbe, Nyanda Elias Ntinginya, Lucas Maganga.

**Data curation:** Willyhelmina Olomi, Wiston William, Nyanda Elias Ntinginya, Lucas Maganga.

**Formal analysis:** Amani Kway, Nyanda Elias Ntinginya, Lucas Maganga.

**Funding acquisition:** Amani Kway, Issa Sabi, Ruby Doryn Mcharo, Nyanda Elias Ntinginya, Lucas Maganga.

**Investigation:** Amani Kway, Ruby Doryn Mcharo, Erica Sanga, Nyanda Elias Ntinginya, Lucas Maganga.

**Methodology:** Amani Kway, Issa Sabi, Willyhelmina Olomi, Ruby Doryn Mcharo, Erica Sanga, Wiston William, Ombeni Chimbe, Nyanda Elias Ntinginya, Lucas Maganga.

**Project administration:** Amani Kway, Ruby Doryn Mcharo, Nyanda Elias Ntinginya, Lucas Maganga.

**Resources:** Amani Kway, Nyanda Elias Ntinginya, Lucas Maganga.

**Software:** Nyanda Elias Ntinginya, Lucas Maganga.

**Supervision:** Amani Kway, Erica Sanga, Nyanda Elias Ntinginya, Lucas Maganga.

**Validation:** Willyhelmina Olomi, Nyanda Elias Ntinginya, Lucas Maganga.

**Visualization:** Amani Kway, Willyhelmina Olomi, Ruby Doryn Mcharo, Erica Sanga, Nyanda Elias Ntinginya, Lucas Maganga.

**Writing – original draft:** Amani Kway.

**Writing – review & editing:** Amani Kway, Issa Sabi, Willyhelmina Olomi, Ruby Doryn Mcharo, Erica Sanga, Ombeni Chimbe, Nyanda Elias Ntinginya, Lucas Maganga.

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
