## [Decision Letter · Decision Letter 0]

6 Dec 2021

PGPH-D-21-00764

HIV testing and linkage to care - A case of a mobile diagnostic and counseling service in Mbeya, Tanzania; A quantitative study

Dear Dr. Kway,

Thank you for submitting your manuscript to PLOS Global Public Health. After careful consideration, we feel that it has merit but does not fully meet PLOS Global Public Health’s publication criteria as it currently stands. Therefore, we invite you to submit a revised version of the manuscript that addresses the points raised during the review process.

We look forward to receiving your revised manuscript.

Kind regards,

Lei Gao

Academic Editor

Journal Requirements:

1. Please update the completed 'Competing Interests' statement, including any COIs declared by your co-authors. If you have no competing interests to declare, please state "The authors have declared that no competing interests exist". Otherwise please declare all competing interests beginning with the statement "I have read the journal's policy and the authors of this manuscript have the following competing interests:"

2. In the online submission form, you indicated that "The datasets used and/or analyzed during the current study are available from the corresponding author on reasonable request". 

3. Please amend your detailed Financial Disclosure statement. This is published with the article, therefore should be completed in full sentences and contain the exact wording you wish to be published.

ii). State the initials, alongside each funding source, of each author to receive each grant.

iii). State what role the funders took in the study. If the funders had no role in your study, please state: “The funders had no role in study design, data collection and analysis, decision to publish, or preparation of the manuscript.”

Additional Editor Comments (if provided):

Reviewers' comments:

Reviewer's Responses to Questions

**Comments to the Author**

1. Does this manuscript meet PLOS Global Public Health’s publication criteria? Is the manuscript technically sound, and do the data support the conclusions? The manuscript must describe methodologically and ethically rigorous research with conclusions that are appropriately drawn based on the data presented.

Reviewer #1: Yes

Reviewer #2: Yes

Reviewer #3: Yes

Reviewer #4: Partly

Reviewer #5: Partly

2. Has the statistical analysis been performed appropriately and rigorously?

Reviewer #1: Yes

Reviewer #2: Yes

Reviewer #3: I don't know

Reviewer #4: No

Reviewer #5: Yes

3. Have the authors made all data underlying the findings in their manuscript fully available (please refer to the Data Availability Statement at the start of the manuscript PDF file)?

Reviewer #1: Yes

Reviewer #2: Yes

Reviewer #3: Yes

Reviewer #4: Yes

Reviewer #5: Yes

4. Is the manuscript presented in an intelligible fashion and written in standard English?

Reviewer #1: Yes

Reviewer #2: Yes

Reviewer #3: Yes

Reviewer #4: Yes

Reviewer #5: Yes

5. Review Comments to the Author

Reviewer #1: This is an interesting well written study from Tanzania. I enjoyed reading the paper. i have a few questions and suggestions.

how do the authors reconcile "In Tanzania 52.2% of PLHIV 15 to 64 years, were aware of their HIV status and 90.9%

75 of those who knew their status were on HAART" mentionned in the introduction with their results should poor linkage to care.

the discussion discusses the present results but if i understand clearly, the 90.9 % on care in Tanzania may reflect the majority of urban populations were care may be more accessible (i am speculating here). Perhaps this conundrum should be discussed is it a rural versus urban contrast? is it border areas? i think such a difference in the same country warrants more discussion.

The title of table 1 should specify the population described for the sake of table autonomy

table 2 on the right the column header anources total n (%) but there are only totals

Reviewer #2: This study assessed the performance of mobile HIV Testing and Counseling (mHTC) services in provision of HIV-testing and linkage to care of newly diagnosed PLHIV from Key and Vulnerable Populations (KVPs. This is a critical issue in Africa as linkage to ART service among PLHIV is of paramount importance to reach epidemic control. There is great need o establish why linkage to ART services is low and to also propose solutions to improve it. This article has thoroughly investigated the issue and identified some issues causing the low linkage.

Reviewer #3: This effort by the authors is a welcome contribution on the subject of linkage to HIV care from the perspective of enhancing uptake of HIV testing through new technologies and more user-friendly platforms in Tanzania and other countries with generalized HIV epidemics. The findings from the study were clear to me and I found them easy to follow. The abstract was well-written.

Major comments:

a) Introduction: There is insufficient literature review of like-minded studies to properly contextualize this study and establish an entry point for the authors. Please bolster your literature review credentials in an additional short paragraph.

b) The data analysis section is woefully under-specified. Please add more detail on the variables assessed (even at the risk of repetition). As it currently stands, the data analysis procedures and parameters are scanty indeed.

c) Discussion: A similar challenge here is insufficient juxtaposition of the authors' own findings with those from other authors say in Uganda, Nigeria, South Africa or Zimbabwe. Although they refer to some previous efforts by other authors, this is effort could be further strengthened to better link the authors findings to those of their peers in the same line of research.

Minor comments:

a) Please reduce the use of brackets in your introduction and endevour to use free flowing text as much as possible. A case in point are lines 59-60.

b) Line 69: I would write.. Despite massive investments...

c) Methods: The study design is not explicitly stated. There are multiple sub-headings lumped together that begin this chapter which appears inappropriate.

Reviewer #4: Comment on PGPH-D-21-00487, Kway et al.,

This interesting paper by Kway et al explores the performance of mobile HIV testing and counseling services and linkage to care for newly diagnosed key and vulnerable populations. Linkage to care is an essential part of HIV-care targets and is a necessity for achieving the UNAIDS 90-90-90 targets. This paper focuses its analysis on key and vulnerable populations which are a known risk group for poor linkage and retention to care making this a relevant and needed analysis. There are comments listed below needed to be addressed prior to publication:

- The methods section needs to be revised to include more details about the study approach. Information on the mHTC data needs to be clear. What type of data/variables are available from mHTC? How does it work in this population? Where are patients referred for linkage using mHTC? The population also needs to be clearly defined, a more specific definition of KVPs is needed. The following sentence “High population mobility associated with cross-border socioeconomic activities, KVPs such as Female Sex Workers (FSWs), miners, long distance truck drivers, poverty and few health facilities near to target populations are 94 among the important factors driving the HIV epidemic in Mbeya” is more of a background statement than method. More information on the catchment of Mbeya, how many clinics are serviced, how big the population is, who the KVP population is needed.

- Linkage needs to be clearly defined. What time-period is this applicable? A confirmation of HIV diagnosis to first HIV-clinic visit is very vague and leaves room for uncertainty on the time-frame in which linkage can occur. In the era of test and treat a 2-week delay in linkage is substantial and it is unclear in the methods what period of time was allowed from first diagnosis to linkage.

- Data collection section needs major revision. It is unclear if the 25,248 records were only from KVPs as exclusion only mentions those who were aware of their status prior to mHTC. How did they determine if someone already knew their HIV status prior to mHTC? Again, what does mHTC mean in this context? Text message for appointment? It needs to be absolutely clear to the reader what mHTC involves.

- Data analysis section is weak. Sentence stating decision to categorize variables was based on previous studies but with no references cited in text. More details are needed on the outcome and exposure variables. Assuming linkage was binary (yes/no) based on their definition above, again it is not clear whether adjustment was done for calendar year, time, facility among other key variables. The authors use log-binomial regression but it is unclear what the linear predictor in their model is. Why did they choose log-binomial?

- The authors do well to show table 1 and 2 with socio-demographic factors but it is unclear how they selected these. Are there other available variables? The associations seen in the table may also be due to the number of various factors. Women more likely to be positive is ambiguous as it is unclear what proportion of women in the cohort were sex workers a known high risk group. A variable for type of key population (sex worker, truck driver, fisherman etc.) is necessary.

- The following sentence is unclear- Descriptive analyses revealed that of the diagnosed HIV-positive clients linked to-care 136 5/6 (83.33%) were children below 15 years, 238/450 (52.89%) were females, 39/64 (60.94%) 137 were widows. Is there a reason why authors highlight widows? What is the rationale for marital status to be associated to linkage? The inclusion of children (younger than 15) makes this analysis more complex to interpret. The association that those younger than 15 are more likely to link is an entirely different question as pediatric and adolescent HIV programs have their own clinics, practices, approaches when compared to adults. The small sample size 5/6 children linking could also be overestimating this association.

- The predictors to linkage of care (table 3) is not clear. Would rather state factors associated with linkage to care not predictors especially in this retrospective approach. The lack of inclusion of other key variables makes this analysis weak. A strength would be including treatment variables and some longitudinal data since the analysis is based on data from an electronic medical database. Linkage could be better defined and even care engagement and retention could be explored. There are no strong conclusions from this analysis and this makes the overall message of the paper weak. Lack of data on the clinics patient linked to care, how they linked, when they linked make the analysis very hard to interpret. Major revisions are needed here.

Minor comments:

- The introduction could be strengthened by incorporating a few sentences on KVPs and what linkage to care, including the 90-90-90 cascade looks like in this population and context. Statistics are briefly mentioned on these targets in the general 15-64y population; however, more effort could be made to include context specific challenges to linkage to care. A more detailed explanation of mobile health testing and counseling could also be introduced in the background.

Reviewer #5: Methods

Mention the parameters of interest

Define the older age group, young children, adults

Results

The majority of the clients were in the age group 25-45 51.71% (put the percentage in brackets). It's easier to read.

Table 1: put the n. Complete the title: socio-demographic characteristics of the study population in general

Not categorized: young children (to be defined) and mention that it is the same group as not applicable.

Gender and sex used interchangeably.

Table 2: fourth column: Total n (%): there is no percentage.

Under table 2: give the explanation of non applicable.

Discussion

Too many results included in this part although they have already mentioned in the Results section.

In the discussion on the frequencies of seropositivity in relation to divorced clients and widowers: this is explained by the fact that the majority were women. It is not shown in the results.

In the conclusion, present the highlights. No longer resume results and do not refer to THIS 2016-2017. Give the conclusions of your work.

6. PLOS authors have the option to publish the peer review history of their article (what does this mean?). If published, this will include your full peer review and any attached files.

**Do you want your identity to be public for this peer review?** For information about this choice, including consent withdrawal, please see our Privacy Policy.

Reviewer #1: No

Reviewer #2: No

Reviewer #3: No

Reviewer #4: No

Reviewer #5: No

---

## [Decision Letter · Decision Letter 1]

13 Apr 2022

HIV testing and linkage to care - A case of a mobile diagnostic and counseling service in Mbeya, Tanzania; A quantitative study

PGPH-D-21-00764R1

Dear Dr. Kway,

We are pleased to inform you that your manuscript 'HIV testing and linkage to care - A case of a mobile diagnostic and counseling service in Mbeya, Tanzania; A quantitative study' has been provisionally accepted for publication in PLOS Global Public Health.

Best regards,

Lei Gao

Academic Editor

Reviewer Comments (if any, and for reference):

Reviewer's Responses to Questions

**Comments to the Author**

1. If the authors have adequately addressed your comments raised in a previous round of review and you feel that this manuscript is now acceptable for publication, you may indicate that here to bypass the “Comments to the Author” section, enter your conflict of interest statement in the “Confidential to Editor” section, and submit your "Accept" recommendation.

Reviewer #1: All comments have been addressed

2. Does this manuscript meet PLOS Global Public Health’s publication criteria? Is the manuscript technically sound, and do the data support the conclusions? The manuscript must describe methodologically and ethically rigorous research with conclusions that are appropriately drawn based on the data presented.

Reviewer #1: Yes

3. Has the statistical analysis been performed appropriately and rigorously?

Reviewer #1: Yes

4. Have the authors made all data underlying the findings in their manuscript fully available (please refer to the Data Availability Statement at the start of the manuscript PDF file)?

Reviewer #1: Yes

5. Is the manuscript presented in an intelligible fashion and written in standard English?

Reviewer #1: Yes

6. Review Comments to the Author

Reviewer #1: (No Response)

7. PLOS authors have the option to publish the peer review history of their article (what does this mean?). If published, this will include your full peer review and any attached files.

**Do you want your identity to be public for this peer review?** For information about this choice, including consent withdrawal, please see our Privacy Policy.

Reviewer #1: No
